# REVISITING THE LOTTERY TICKET HYPOTHESIS: A RAMANUJAN GRAPH PERSPECTIVE

## ABSTRACT

Neural networks for machine learning applications often yield to weight pruning resulting in a sparse subnetwork that is adequate for a given task. Retraining these 'lottery ticket' subnetworks from their initialization minimizes the computational burden while preserving the test set accuracy of the original network. The existing literature only confirms that pruning is needed and it can be achieved up to a certain sparsity. We analyze the pruned network in the context of the properties of Ramanujan expander graphs. We consider the feed-forward network (both multi-layer perceptron and convolutional network) as a series of bipartite graphs which establish the connection from input to output. Now, as the fraction of remaining weights reduce with increasingly aggressive pruning, distinct regimes are observed: initially, no significant decrease in accuracy is demonstrated, and then the accuracy starts dropping rapidly. We empirically show that in the first regime, the pruned lottery ticket sub-network remains a Ramanujan graph. Subsequently, with the loss of Ramanujan graph property, accuracy begins to reduce sharply. This characterizes an absence of resilient connectivity in the pruned sub-network. We also propose a modified iterative pruning algorithm which removes edges in only the layers that are Ramanujan graphs thus preserving global connectivity even for heavily pruned networks. We perform experiments on MNIST and CI-FAR10 datasets using different established feed-forward architectures to support the criteria for obtaining the winning ticket using the proposed algorithm.

## 1 INTRODUCTION AND RELATED WORK

*Neural Network (NN)* and its recent advancements have made a significant contribution to solve various machine learning applications. The power of an over-parameterized NN lies in its capability to learn simple patterns and memorize the noise in the data (Neyshabur et al., 2018). However, the training of such networks requires enormous computational resources, and often the deployment onto low-resource environments such as mobile devices, or embedded systems becomes difficult. Recent trend in research to reduce training time of deep neural networks has shifted towards pre-training following the introduction of a remarkable contribution, named the *Lottery Ticket Hypothesis (LTH)*, which hypothesize the existence of a highly sparse subnetwork and weight initialization to reduce the training resources as well (Frankle & Carbin, 2019). It uses a simple iterative, magnitude-based pruning algorithm, and empirically shows that even after removing approximately 90% of the weights, the subnetwork can preserve the original generalization error. In the subsequent studies, the focus goes on finding this lottery ticket for more competitive tasks by pruning with weight rewinding(Frankle et al., 2019a), fine tuning the learning rates (Renda et al., 2020), more efficient training (You et al., 2019; Brix et al., 2020; Girish et al., 2021).

Neural network pruning involves sparsification of the network (LeCun et al., 1990; Blalock et al., 2020). It identifies the weight parameters, removal of which incurs minimal effect on the generalization error. There exists different categories of pruning based on (i) how the pruning is performed, for instance based on the weight magnitude (Han et al., 2015; Zhu & Gupta, 2017), gradient in the backpropagation, hessian of the weight (Hassibi et al., 1993; Dong et al., 2017; Lee et al., 2018), etc; (ii) whether the pruning is global or local; (iii) how often pruning should be applied like one-shot Lee et al. (2018); Wang et al. (2020), iterative Tanaka et al. (2020). One of the primary goals in the literature has been to reduce the computational footprint at the time of prediction, i.e., during post-training. In recent LTH studies, the victim weights are determined by their value at the ini-

tialization, gradient of the error, and network topology Lee et al. (2019); Tanaka et al. (2020). To understand weight initialization, Malach et al. (2020) show that pruning makes a stronger hypothesis with bounded weight distribution. The sparsity of the network is reduced from polynomial to a logarithmic factor of the number of training variables (Orseau et al., 2020). Mocanu et al. (2018) suggest to consider the topology of the network from a network science point of view. The pruning algorithm starts from a random *Erdős Rényi* graph and returns a scale-free network of a high sparsity factor based on the number of neurons in each layer. The method is further evolved for convolution layers considering both the magnitude and gradient of the weights(Evci et al., 2020a).

Various analysis for explaining the LTH have been attempted in the past. Researchers Evci et al. (2020b) explain empirically why the LTH works through gradient flow at different stages of the training. Despite previous attempts to explain why the Lottery Ticket Hypothesis works, the underlying phenomenon associated with the hypothesis still remains ill-understood. All of these studies related to LTH identify that a sparse sub-network can be trained instead of a complete network and the network needs to be connected from input to output layers. However, none of them try to explain the LTH and the properties of the pruned network through the lens of spectral graph theory. The network connectivity can be described from the *graph expansion* point of view, where any subset of vertices of size less than or equal to half of the number of vertices in a graph, is adjacent to at least a fraction of the number of vertices in that set; for details, see (Lubotzky, 2010). Graphs satisfying this property are known as expander graphs. The *Ramanujan Graph* is a special graph in a bounded degree expander family, where the eigenbound is maximal (Nilli, 1991). This leads to a maximum possible sparsity of a network while preserving the connectivity.

In this paper, we initiate a study to observe the characteristics of a pruned sub-network from the spectral properties of its adjacency matrix, which, has not been reported previously. We represent a feed-forward neural network as a series of connected bipartite graphs. Both weighted and unweighted bi-adjacency matrices are considered. The Ramanujan graph properties of each of the bipartite layers are studied. We use the results of Hoory (2005) on the bound of spectral gap for the weight matrix of a pruned network. It is empirically observed that existence of winning tickets in a pruned network is dependent on the Ramanujan graph properties of the bipartite layers. As network sparsity increases with more aggressive pruning, we obtain regions where test set accuracy do not decrease much and the bipartite layers satisfy Ramanujan graph property. Subsequently we obtain regions where the Ramanujan graph properties are lost for all the layers and test accuracy decreases sharply. Also, we observe that the impact of noise in the data, on test set accuracy is more adverse when some of the layers lose their Ramanujan graph properties. Experimental results are presented for the Lenet architecture on the MNIST dataset and the Conv4 architecture on the CIFAR10 dataset. Results for other popular feed-forward network are presented in the Appendix.

We suggest that preservation of Ramanujan graph properties may benefit existing network pruning algorithms. We propose a modified pruning algorithm that uses the spectral bounds. The algorithm identifies network layers that may still be pruned further, while avoiding pruning in layers that have already lost their Ramanujan graph property. Neural network weight score functions suggested by various pruning algorithms are used to represent the bipartite layers as weighted graphs. Spectral bounds for these graphs are used to verify the Ramanujan property. For a number of popular pruning algorithms, we experimentally demonstrate significant improvement in accuracy for sparse networks by using these connectivity criteria.

**Contributions:** The contributions of the paper are the followings:
1. We propose a methodology for analyzing winning lottery tickets with respect to their spectral properties.
2. We empirically observe that winning lottery tickets often satisfy layerwise bipartite Ramanujan graph property representing a sparse but resiliently connected global network. The property is checked using spectral bounds that generalize to irregular networks. We also notice better noise robustness when all the layers of the pruned sparse networks preserve the Ramanujan graph property.
3. Based on the above property we propose a modified iterative network pruning algorithm that attempts to preserve Ramanujan graph property for all the layers even at low network densities. It identifies layers that are still amenable to pruning while avoiding further pruning in layers that have lost their Ramanujan graph property. We report significant performance improvement for a number of popular pruning algorithms modified using this criteria.

## 2 THE LOTTERY TICKET HYPOTHESIS AND NETWORK PRUNING

The lottery ticket hypothesis states that a randomly initialized, dense neural network contains a subnetwork which when trained independently using the same initialization achieves a test accuracy close to the original network after training for less or at most the same number of iterations Frankle & Carbin (2019). These subnetworks, denoted as 'winning tickets', can be uncovered by network pruning algorithms. Weight pruning is one such simple strategy. Let the original dense network be represented as the function $\mathcal{N}(x; \theta)$, where $x$ is the input and $\theta$ are the weights. The weights have an initialization of $\theta_0$. Weight pruning generates a mask $m \in \{0, 1\}^{|\theta|}$ such that the pruned network can be represented by $\mathcal{N}(x; m \odot \theta)$ with initialization $m \odot \theta_0$.

Pruning algorithms that are used to obtain the winning tickets can be either one-shot or iterative. In one-shot pruning the original network is trained to convergence, then $p\%$ of weights are pruned and the surviving weights are re-initialized to their values in $\theta_0$ followed retraining/fine tuning the subnetwork. Here, the network training and pruning are not simultaneously performed and pruning occurs only after convergence is reached. Iterative pruning repeats one-shot pruning over several iteration. This often leads to higher pruning percentage while retaining test set accuracy. However, iterative pruning is more time consuming than one-shot pruning. After pruning the surviving weights may alternately be initialized to their values in $\theta_i$, $i$ denoting a small iteration number, rather than their initial values in $\theta_0$. This process, as illustrated in Figure 1 (Frankle et al., 2019b), is denoted as rewinding and is effective for large networks. We adopt this in our study.

Various scoring functions are used to prioritize the weights for pruning. They may be based on weight magnitudes, gradient, information flow (Blalock et al., 2020; Hoefler et al., 2021) or saliency Tanaka et al. (2020). Magnitude pruning provides a simple method for obtaining the pruning mask by retaining the top $p\%$ weights $w_i \in \theta$ that have the highest values of $|w_i|$. The role of the weights in local computation in the network layers is not considered. A higher pruning efficiency may be achieved by algorithms that account for connectivity structures in the individual layers. In the next section we describe graph parameters of the network that determine such connectivity.

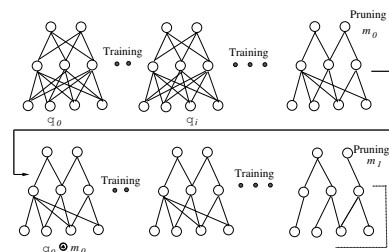

Figure 1: Iterative Network Pruning with Rewinding Frankle et al. (2019b)

## 3 EXPANDERS AND RAMANUJAN GRAPHS

Expanders are highly connected, and yet sparse graphs. In this work, we shall be considering finite, connected, undirected, but not necessarily regular graphs. Recall that the degree of a vertex $v$ in a graph is the number of half edges emanating from $v$.

**Definition 1 ($(n, d, \varepsilon)$-expander)** *Let $\varepsilon > 0$. An $(n, d, \varepsilon)$-expander is a graph $\mathcal{G} = (V, E)$ on $|V| = n$ vertices, having maximal degree $d$, such that for every set $\emptyset \neq U \subseteq V$ satisfying $|U| \leq \frac{n}{2}$, $|\delta(U)| \geq \varepsilon|U|$ holds.*

Here, $\delta(U)$ denotes the vertex boundary of $U$. The quantity $\frac{|\delta(U)|}{|U|}$ measures the rate of expansion and the infimum $\frac{|\delta(U)|}{|U|}$ as $U$ varies among the non-empty subsets of $V$ with $|U| \leq \frac{|V|}{2}$ is called the (vertex) Cheeger constant $h(\mathcal{G})$ of the graph $\mathcal{G}$. The higher the value of $h(\mathcal{G})$, the more expansion property it exhibits and vice versa. Expansion and the Cheeger constant quantifies the connectivity properties of a graph as a high value of $h(\mathcal{G})$ signifies that the graph is strongly connected. This ensures that information can flow freely without much bottlenecks.

**Definition 2 (Expander family)** *A sequence of finite, connected graphs $\{\mathcal{G}_i = (V_i, E_i)\}_{i=1,2,\cdots}$ on $V_i$ vertices and $E_i$ edges is called an expander family if there exists an uniform $\epsilon > 0$ such that each graph in the sequence is an $(|V_i|, d_i, \epsilon)$ expander for some $d_i$'s.*

The study of expansion properties of graphs is closely related to the study of the spectrum (distribution of eigenvalues) of the adjacency operator defined on them. Given a finite $r$-regular

graph of size $|V| = n$, the eigenvalues $t_i$ of the adjacency matrix are all real and they satisfy, $-r \le t_n \le \ldots \le t_2 \le t_1 = r$. The graph is connected iff $t_2 < t_1$ and is bipartite iff $t_i$'s are symmetric about 0 (in particular $t_n = -r$). The quantity $t_1 - t_2$ is known as the *spectral gap* and it is related to the Cheeger constant via the discrete Cheeger-Buser inequality, discovered independently by Dodziuk (1984) and by Alon & Milman (1985). In our context, we consider a stronger notion of the spectral gap (but it is equivalent for bipartite graphs). Let $t := \max\{|t_i| : 1 < i \le n, |t_i| < t_1\}$. In here, the quantity $t_1 - t$ will denote the spectral gap.

The more connected a graph is, the larger is the spectral gap and ideally, a graph with strong expansion properties has a very large spectral gap. However, for a bounded degree expander family, this spectral gap cannot be arbitrarily large. This brings us to the notion of Ramanujan graphs.

**Definition 3 (Ramanujan graph)** *Let $\mathcal{G}$ be a $r$-regular graph on $n$ vertices, with adjacency eigenvalues $\{t_i\}_{i=1,2,\ldots n}$, satisfying $-r \le t_n \le \ldots \le t_2 \le t_1 = r$. Let $t(\mathcal{G}) := \max\{|t_i| : 1 < i \le n, |t_i| < t_1\}$. Then $\mathcal{G}$ is a Ramanujan graph if $t(\mathcal{G}) \le 2\sqrt{r-1} = 2\sqrt{t_1 - 1}$.*

The fact that in a bounded degree expander family, the eigenvalue bound in Ramanujan graphs is maximal can be deduced from the following result due to Alon, see Nilli (1991), $t(\mathcal{G}) \ge 2\sqrt{r-1} - \frac{2\sqrt{r-1}-1}{\lfloor m/2 \rfloor}$ where $m$ denotes the diameter of the graph. As $m \to +\infty$ and $r$ is bounded, we obtain the result (this also follows from the Alon-Boppana theorem).

For our applications, we shall encounter not necessarily regular graphs, thus we need a notion of irregular version of Ramanujan graphs. The two ways we shall exploit Definition 3 for irregular graphs will be to:

1. Use the average degree $d_{avg}$ in place of the regularity.
2. For weighted graphs, use $t_1$, the largest eigenvalue of the adjacency matrix.

Note that a motivation for considering the above bounds comes from the following generalisation of the definition of Ramanujan graphs to irregular graphs. For a finite, connected graph $\mathcal{G}$ (not necessarily regular) consider its universal cover $\tilde{\mathcal{G}}$, for details see (Hoory et al., 2006, sec. 6). A Ramanujan graph is a graph satisfying $t(\mathcal{G}) \le \rho(\tilde{\mathcal{G}})$ where $\rho(\tilde{\mathcal{G}})$ denotes the spectral radius of $\tilde{\mathcal{G}}$. See also Marcus–Spielman–Srivastava (Marcus et al., 2015, sec. 2.3). A result of Hoory, see Hoory (2005) implies that $2\sqrt{d_{avg} - 1} \le \rho(\tilde{\mathcal{G}})$. Thus, it makes sense to consider $t(\mathcal{G}) \le 2\sqrt{d_{avg} - 1} \le \rho(\tilde{\mathcal{G}})$. The above consideration also result in extremal families, Hoory (2005). Further, for irregular bipartite graphs, with minimal degree at least two and an average degree $d_{avgL}$ on the left and $d_{avgR}$ on the right, we can further consider the sharper estimate $t(\mathcal{G}) \le \sqrt{d_{avgL} - 1} + \sqrt{d_{avgR} - 1} \le \rho(\tilde{\mathcal{G}})$ see Hoory (2005).

Upto now, we had only discussed about unweighted graphs. A weighted graph is a graph with a weight function $w : E \to \mathbf{R}_{\ge 0}$ attached on the edges. It can be looked upon as a generalisation of unweighted graphs, in the sense that in the unweighted case, the function $w$ takes values in the set $\{0, 1\}$. In the case of weighted networks, we shall use the absolute values of the weight functions according to the architecture for the corresponding dataset. This also means that in the case of weighted graphs, we need to modify the definition of the edge set of the graph to incorporate multiple (as well as fractional) edges. The theory of characterisation of weighted Ramanujan graphs is not well developed. However, characterisation of weighted expanders (with positive weights) exist, due to the Cheeger inequality for such graphs, see (Chung, 1996, sec. 5) and we use the largest eigenvalue of the adjacency matrix in place of the regularity. In the case of regular graphs, it coincides with the notion of Ramanujan graphs and even in the general case, by the Cheeger inequality, it ensures a large expansion which in turn implies that there is no bottleneck to the free flow of information. This forms the theoretical basis of our work.

## 4 RAMANUJAN GRAPH CHARACTERIZATION OF NEURAL NETWORKS

We can represent any neural network using graphs (possibly weighted, depending on the context) and in this article, we will be dealing with a sequence of finite, bipartite graphs. This is because if $\mathbf{N}$ denotes a neural network having $l$ layers $\mathbf{N}_1, \mathbf{N}_2, \ldots, \mathbf{N}_l$ respectively, then each $\mathbf{N}_i$ ($i = 1, 2, \ldots, l$)

is a complete bipartite graph to start with. A pruned subnetwork results in edge sparsification of the underlying graphs. For that purpose, we need to approximate complete, bipartite graphs by sparse, proper subgraphs. The motivation to study pruning based on expander characteristics stems from the fact that complete graphs can be approximated using expanders, see Spielman (2018). The notion of Ramanujan graphs allow us to quantify the pruning limit, and we empirically justify our technique.

In Figure 2, we present examples of three small feed-forward networks with a single hidden layer and four neurons in each layer. The networks consists of two bipartite graphs corresponding to the input-hidden and hidden-output layers. The sparsity of all the three networks, as measured by the number of edges present, are the same. However, in the first network none of the layer-wise bipartite graphs satisfy the expander property and thus information flow from a significant number of input nodes to output nodes are disrupted. For the second network, the input-hidden layer bipartite graph is an expander, while the hidden-output layer bipartite graph is not an expander.

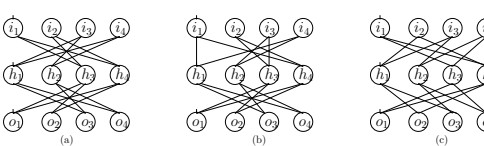

Figure 2: Examples of small feed-forward networks having same sparsity but different connectivity properties; (a) disconnected network, (b) partially connected (c) strongly connected. When all of them are scaled to larger networks, (c) has large spectral gap compared to (a) and (b) signifying a high rate of expansion.

Here few of the flow paths from input to the output nodes are disconnected. Both the bipartite graphs for the third network are expanders, thus all the inputs nodes are connected to all the output nodes. We denote the first network as a disconnected one, the second network as partially connected, and the third one as fully connected. The example illustrates that layer-wise sparse bipartite expander graphs ensures global information flow across layers. While connected but non-expander sparse bipartite layers do not necessarily lead to global connectivity across layers. It is known in literature that sparse but resiliently connected neural networks not only have good generalization performance but also achieve noise robustness (Liu et al., 2018).

## 4.1 BIPARTITE GRAPH STRUCTURE

In this work, we focus on the fully-connected layers, and convolution layers of the feed-forward neural network only. Since, pruning is a part of network compression, we only consider the trainable layers here. We consider both unweighted and weighted representations of the bipartite graphs. We ignore signs of the weight values and consider only the magnitudes for the weighted graph representation. Even though sign of the weights are important for determining the neural network functionality, we argue that for studying their connectivity properties only the magnitudes need to be considered.

**Fully Connected Layers (FC)**: For a fully connected layer $\mathbf{N}_i$, with $n^{i-1}$ number of inputs and $n^i$ number of outputs the weighted bi-adjacency matrix of the corresponding graph is $\mathbf{W}_i \in \mathbb{R}^{[n^{i-1} \times n^i]}$, and the corresponding pruning mask is $\mathbf{M}_i \in \{0, 1\}^{[n^{i-1} \times n^i]}$.

**Convolution Layers (Conv)**: Here, we consider the kernel size, the number of input and output channels to unfold the layer into a complete bipartite graph. For a convolution layer $\mathbf{N}_i$ with the kernel of size $k$, $n^{i-1}$ input, and $n^i$ output channels, the weighted bi-adjacency matrix of the corresponding graph will be $\mathbf{W}_i \in \mathbb{R}^{[(n^{i-1}.k.k) \times n^i]}$, and the corresponding pruning mask is $\mathbf{M}_i \in \{0, 1\}^{[(n^{i-1}.k.k) \times n^i]}$. An example is shown in Figure 3.

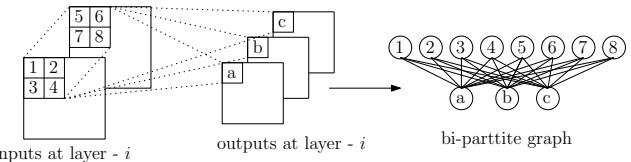

Figure 3: An example of bipartite graph computation for a particular convolution layer with kernel size $2 \times 2$, 2 input channels, and 3 output channels

Now, the bounds are analyzed for both unweighted ($\mathbf{M_i}$) and weighted ($\mathbf{W}_i$) bipartite graphs at each layer $\mathbf{N}_i$. Table 1 describes different bounding constraint depending on the type of the considered graph and bound type. We consider the bound differences ($\Delta_S$ and $\Delta_R$) for the eigenvalue and average degree respectively. A transition of the $\Delta$ values from positive to negative denotes a violation of the bounds, and thus loss of Ramanujan graph property of the bipartite graph for the corresponding pair of layers in the feed-forward neural network.

Table 1: Different bound criteria on the second largest eigenvalue $t_2$ of the bi-adjacency matrices

| BI-ADJACENCY | EIGENVALUE (eb) | AVERAGE DEGREE (db) |
|---|---|---|
| Unweighted ($\mathbf{M}_i$) | UE: $t_2(\mathbf{M}_i) \leq 2\sqrt{t_1(\mathbf{M}_i) - 1}$ | UD: $t_2(\mathbf{M}_i) \leq \sqrt{d_{avgL}(\mathbf{M}_i) - 1} + \sqrt{d_{avgR}(\mathbf{M}_i) - 1}$ |
| Weighted ($\mathbf{W}_i$) | WE: $t_2(\mathbf{W}_i) \leq 2\sqrt{t_1(\mathbf{W}_i) - 1}$ | |
| difference on bound | $\Delta_S = (2\sqrt{t_1 - 1} - t_2)/t_2$ | $\Delta_R = (\sqrt{d_{avgL} - 1} + \sqrt{d_{avgR} - 1} - t_2)/t_2$ |

## 4.2 RAMANUJAN GRAPH PROPERTY PRESERVING PRUNING ALGORITHM

In this section, we describe a modification of the iterative pruning algorithm that preserves the Ramanujan graph property of a pruned neural network. In iterative pruning algorithms, pruning is employed once the training is complete (i.e. after it converged to an optimum or it reached certain epochs). The pruned network is trained again with its initial weight values to perform the next pruning iteration. The magnitude of a score function $\theta$ is used to identify the victim weights. The weights are made to zero if their score function values lie in the bottom $p$ percentile. In the Iterative Magnitude Pruning (IMP) Frankle & Carbin (2019), magnitude of the weight is used as the score function. More sophisticated score functions are used in SynFlow (Tanaka et al., 2020), SNIP (Lee et al., 2018).

---

**Algorithm 1:** LAYER-WISE CONNECTIVITY BASED PRUNING

**Input:** Trained Network $\mathbf{N}$, Score function $\theta$, Pruning percentile $p$, pruning level $K$
**Output:** Pruned Network $\mathbf{N}^\star$

1   $\mathbf{N}^\star \leftarrow \mathbf{N}$
2   **for** $i \leftarrow 1$ **to** $l$ **do**
3     $k \leftarrow 1$
4     **while** $k \leq K$ **do**
5       $\mathbf{N}_i^{temp} \leftarrow$ PRUNEBYPERCENTILE$(p/2^k, \mathbf{N}_i, \theta_i)$    // Prune using the score function
6       CALCULATE $d_{avgL}$ AND $d_{avgR}$ OF $\mathbf{N}_i^{temp}$
7       **if** $min(d_{avgL}, d_{avgR}) < 2)$ **then**
8         $\mathbf{N}_i^\star \leftarrow \mathbf{N}_i$       // To avoid layer collapse
9         BREAK
10       **end**
11       CALCULATE $\Delta_S, \Delta_R$ FOR $\mathbf{N}_i^{temp}$
12       **if** $max(\Delta_S, \Delta_R) > 0$ **then**
13         $\mathbf{N}_i^\star \leftarrow \mathbf{N}_i^{temp}$   // Satisfy Ramanujan Property on the unweighted graph
14         BREAK
15       **else**
16         CALCULATE $\Delta_S^\theta$ FOR $\mathbf{N}_i^{temp}$ USING $\theta_i$
17         **if** $\Delta_S^\theta \geq 0$ *and* $t_1^\theta \geq 1$ **then**
18           $\mathbf{N}_i^\star \leftarrow \mathbf{N}_i^{temp}$   // Satisfy Expansion Property $\mathbf{N}_i$ using $\theta_i$ as weights
19           BREAK
20         **end**
21       **end**
22       $k \leftarrow k + 1$
23     **end**
24   **end**
25   nure **return** $\mathbf{N}^\star$

---

We propose a modification to the above method. If a weight is identified as a victim based on the above score function based scheme; we check the Ramanujan property of the layer $\mathbf{N}_i$ to which this weight belongs. First, we verify the connectivity property of the layer using the bounds on the unweighted graph as defined in Table 1($\Delta_S, \Delta_R > 0$). If the unweighted graph bounds are not satisfied, we consider a weighted graph with score function values as the edge weights. The bounds using the eigenvalues of this weighted graph is used to verify the spectral expansion property ($\Delta_S > 0$) for a particular layer. If the pruned network $\mathbf{N}_i^{temp}$ follows the Ramanujan graph property given by the bounds, then the algorithm simply proceeds to the next level of pruning operation.

If the bound is not satisfied for a particular layer $i$, the weights are simply reset to the values before pruning and the pruning percentile $p$ is halved and search for a better approximation of $\mathbf{N}_i$ is resumed. The target pruning level $k \in [K]$ is a controlling parameter to limit the search iteration.

The proposed algorithm attempts to preserve Ramanujan property for as many layers as possible by using the spectral bound criteria to determine the layers which are still amenable to pruning. It avoids further pruning of the layers that have lost their connectivity property. We experimentally show that the approach is effective when used with IMP, SNIP, SynFlow pruning algorithms with their designated score functions.

The method is described in Algorithm 1. It takes the trained network $\mathbf{N}$, score function $\theta$, pruning percentile $p$, pruning level $K$ as inputs and returns the pruned network $\mathbf{N}^\star$ as output. This algorithm is called at each pruning iteration, and continued till the stopping criterion is met, i.e., $\mathbf{N}_i^\star = \mathbf{N}_i, \forall i$. As discussed in Section 3, we mainly study two types of bounds on the second largest eigenvalue of the bipartite graph $t_2 = t(\mathcal{G})$, i.e., (i) **eb**- based on the largest eigenvalue($t_1$) and (ii) **db**- based on the average degree ($d_{avgL}$ and $d_{avgR}$). Here, $\mathcal{G}$ for a bipartite neural network layer $\mathbf{N}_i$ is either the unweighted graph or a weighted graph with score functions as the weights. In order to avoid 'layer collapse' we stop pruning for a layer if the bipartite graph has a node with degree less than two in either of the parts.

The top-2 largest magnitude eigenvalues at layer $i$ are computed from the symmetric matrix $\mathbf{N}_i^T \mathbf{N}_i$ with implicitly restarted Arnoldi methods (Lehoucq et al., 1998) and, the average degree is computed from the connected component of the bipartite-graph. Hence, the computational complexity is mainly driven by the pruning level $K$, the number of input and output neurons at each layer, and the number of layers. The additional space will be required to store the bipartite graph of a layer. For the convolutional layers the size of the adjacency matrix is effectively same as the size of the convolution kernel which is usually small. Also, the usually the number of fully connected layers are not very high.

## 5 EXPERIMENTAL RESULTS

We have used the MNIST and CIFAR10 datasets in our study. As an evaluation measure we use the classification accuracy for both clean and noisy test sets. Noisy test sets were generated by adding zero mean $\sigma$ variance Gaussian noise to the image pixels. We report results for the Lenet, and Conv4 network architectures following the methodology adopted in Frankle & Carbin (2019). We also perform the studies on other networks which are presented in the Appendix. Hyperparameter values used in our experiments are reported in Table 3 in the Appendix.

### 5.1 EXPERIMENTAL SETUP

We study the variation of classification accuracy with the network density as measured by the remaining weights percentile $100 - p$, where $p$ is the pruning percentile. For different layers the percentile can be different, as denoted by $p_{fc}$ for fully connected (FC) layers, $p_{conv}$ for convolution layers, and $p_{out}$ for the output layers. We study the two cases for the bound specific to unweighted ($\mathbf{M}_i$) and weighted graph ($\mathbf{W}_i$). For each of the studies, we plot four parameters; (i) eigenbound bound difference ($\Delta_S$), (ii) average degree bound difference ($\Delta_R$),(iii) network density for each layer, and (iv) test accuracy on both clean and noisy data. The degree and eigenbound differences are defined in Table 1.

**Relationship between Ramanujan graph properties and LTH**   The primary goal of our study is to identify the LTH regimes, determined by the spectral properties of the bipartite-network layers. The existence of LTH regime is illustrated for two representative established networks. The results for Lenet on MNIST dataset is shown in Figure 4; and for Conv4 on CIFAR10 is shown in Figure 5. In each of the plots, we show the variation of classification accuracy on clean and noisy test sets (having various noise levels ($\sigma$)), with the remaining weights percentile. The spectral properties are characterized by eigenbound difference $\Delta_S$ for both the weighted and unweighted graph representations, and the degree bound difference $\Delta_R$ for only the unweighted graph representation. A transition of $\Delta_S$ and $\Delta_R$ from positive to negative values denotes the loss of Ramanujan graph property of the corresponding layer bipartite graph. Accordingly, the plot is divided into three regimes - (i) fully Ramanujan, where all the layers maintain the Ramanujan property i.e., $\Delta > 0, \forall \mathbf{N}_i$, (ii) partially Ramanujan, where some of the layers maintain the property, and (iii) non-Ramanujan, where none of the layers retain the property. For the Conv4 network which has more number of layers we

show results for the unweighted graph representation only. Similar results for other networks and different pruning settings are discussed in the Appendix.

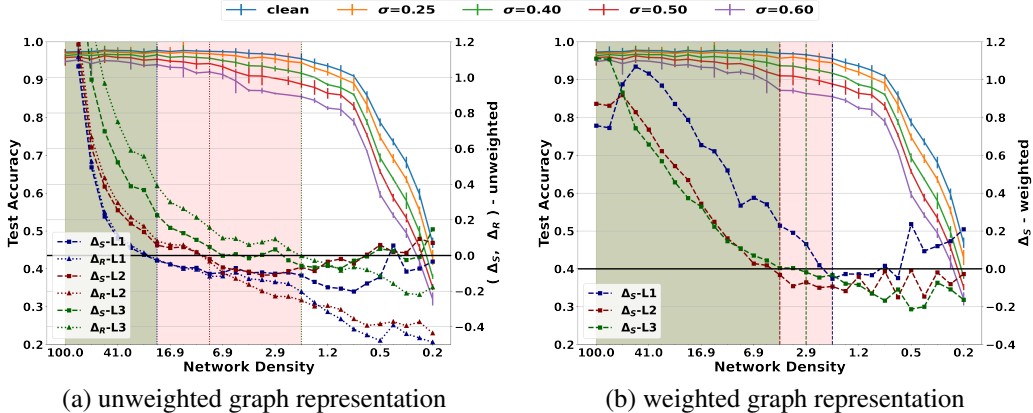

(a) unweighted graph representation

(b) weighted graph representation

Figure 4: Results for MNIST dataset on Lenet architecture; (a) considering unweighted bi-adjacency matrices, (b) considering weighted bi-agjacency matrices. Variation of accuracy with network density is plotted for the clean and noisy test sets with increasing noise variances $\sigma$. Error bars for the accuracy values computed over 5 runs are shown. For the layers L1, L2, and L3 the values of $\Delta_S$ are plotted for both unweighted and weighted representations, and $\Delta_R$ for only the unweighted representation. As mentioned in Table 1, $\Delta_S$ and $\Delta_R$ denote the difference in bounds of the eigenvalues and average degrees. Transition of the $\Delta$ values from positive to negative denote the loss of Ramanujan graph property. The plot is divided into three regimes- *fully Ramanujan* (gray shade), where the Ramanujan graph property holds for all the layers, *partially Ramanujan* (pink shade), where the property holds for some of the layers, and *non-Ramanujan* (no shade) where none of the layers retain the property.

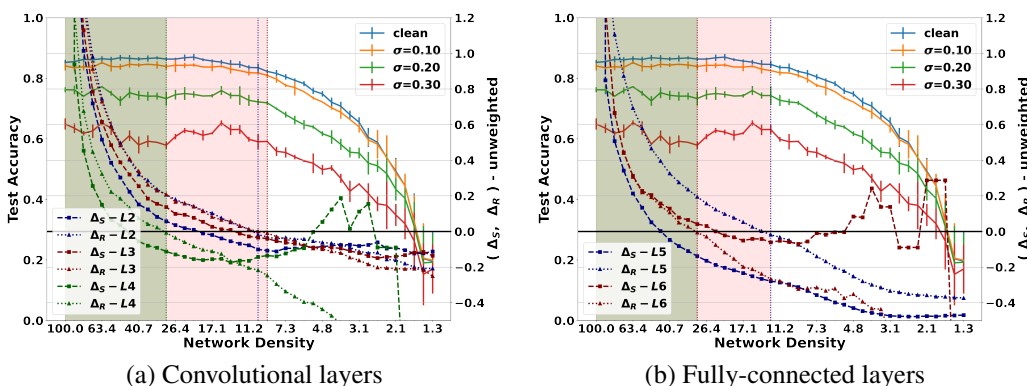

(a) Convolutional layers

(b) Fully-connected layers

Figure 5: Results for CIFAR10 dataset on Conv4 architecture considering unweighted graph. Results for the convolution layers (L2,3,4) is shown in (a), while for the FC layers (L5,6) is shown in (b). We exclude the first and last layers in this study, due to the low cardinality of one of the parts in the bipartite graph for these layers. These layers are usually not pruned by the pruning algorithms.

In all the figures, we observe that the accuracy values start dropping sharply beyond the partially Ramanujan graph property boundary (pink shade) when all the layers loose their Ramanujan graph property. Accuracy also starts reducing slowly from the fully Ramanujan boundary (gray shade). The accuracy on clean data falls sharply when the layers do not satisfy the bounds for the weighted graphs. The bounds for the unweighted graph allow only the robust winning tickets, having high accuracy even on noisy data. For very low network densities it is seen that the $\Delta$ values become positive again. This is a boundary effect owing to the in-applicability of Ramanujan graph bounds for disconnected graphs.

Table 2: Results of different pruning algorithms under the Ramanujan Graph property preservation

| Pruning | | Lenet/MNIST | | | Conv4/CIFAR10 | |
| Algorithm | $\alpha$ | Density | Test Accuracy | $\alpha$ | Density | Test Accuracy |
|---|---|---|---|---|---|---|
| Without Pruning | 0.0 | 100.0 | **97.16** | 0.0 | 100.0 | **85.86** |
| IMP | 1.5 | 3.16 | 93.88 | 1.0 | 10.0 | 81.91 |
| IMP | 2.0 | 1.0 | 45.39 | 2.0 | 1.0 | 10 |
| IMP-Bound | - | **3.6** | **96.74** | - | 10.4 | **81.8** |
| SNIP | 1.5 | 3.16 | 79 | 1.0 | 10.0 | 80.3 |
| SNIP | 2.0 | 1.0 | 49.8 | 2.0 | 1.0 | 64.26 |
| SNIP-Bound | - | 7.64 | 95.41 | - | 10.0 | **79.8** |
| SynFlow | 1.5 | 3.16 | 95.92 | 1.0 | 10.0 | 82.5 |
| SynFlow | 2.0 | 1.0 | 49.11 | 2.0 | 1.0 | 69.2 |
| SynFlow-Bound | - | **1.33** | **93.82** | - | 1.15 | 64.5 |

**Comparison of pruning approaches**   We show that consideration Ramanujan graph properties benefits network pruning algorithms. We consider three popular network pruning algorithms - (i) Iterative Magnitude Pruning (IMP) Frankle & Carbin (2019), (ii) the iterative verion of Single-shot Network Pruning based on Connection Sensitivity (SNIP) Lee et al. (2018), (iii) Synaptic flow based pruning SynFlow Tanaka et al. (2020) and show the results of test accuracy achieved with similar densities with and without the bound condition. To do this experimentation, we choose the a fix compression ratio $\alpha$ to achieve the desired density of the pruned network as $10^{-\alpha} \times 100$, and observe the difference in accuracy due to the network connectivity. The results are presented in Table 2 for Lenet/MNIST and Conv4/CIFAR10. For each of the algorithm we consider two cases - (i) the network is pruned to very low densities using existing algorithms, (ii) network pruning is stopped when there is loss of Ramanujan graph properties (#AlgoName-Bound). The bounds are considered as two types of bounds - first it tries to maintain the bounds on the unweighted graph. Next, if the unweighted graph looses the Ramanujan property it tries to preserve the information flow by considering the spectral bounds for the weighted graphs with score function. We use layer-wise pruning for all the algorithms.

Here, adding the bound criteria guides the pruning algorithm to stop at a density where the accuracy is comparable with its original network. These values are marked in bold in Table 2. The highest improvement is achieved by SynFlow algorithm for the Lenet architecture on MNIST dataset. While the usual SynFlow algorithm results in an accuracy of 49.11% at the density of 1.0, the SynFlow-Bound algorithm has an accuracy of 93.82% for a network density of 1.33. For the Conv4 architecture on CIFAR10, the scope of improvement using the bound criteria is limited since the existing algorithms preserve the unweighted graph connectivity property for most of the layers. We have also studied the spectral properties of all the layers the results are presented in the Appendix.

We can also observe the difference of an arbitrary non-expander graph with an expander one of similar density. As an example, we see the result for Conv4/CIFAR10. With density of 3.16 the IMP achieves accuracy of 60.61% while SynFlow meets 77.18% accuracy. By analyzing the detail it has been found that the Ramanujan property is lost in FC layer-1 only for SynFlow where the same is observed in IMP for both Conv Layer-4 and FC layer-1. Hence, IMP loses more Ramanujan graph property as a whole in the entire network.

## 6   CONCLUSION

In this work, we study the validity of the lottery ticket hypothesis (LTH) based on structural connectivity properties of the neural network. Ramanujan graph properties of the bipartite layers are studied in terms of certain spectral bounds. As test accuracy varies with decreasing network density as a result of pruning, three distinct regions are demarcated using these bounds. In the first region, all the bipartite layers are Ramanujan graphs, in the second region some of them are, and in the third low network density regions none of the layers are Ramanujan graphs. We empirically demonstrate the validity of the lottery ticket hypothesis robustly in the first region and partially in the second region. We propose a modification of existing iterative pruning algorithms that preserves Ramanujan graph property. Further refinement of this approach of pruning can result in more efficient winning ticket search, which will be the basis of future research.

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

# A    APPENDIX

We present additional results to supplement those given in the paper. The results for corresponding to the LTH hypothesis is presented. Then detailed results concerning the pruning algorithms are described.

## A.1    MORE RESULTS FOR LENET ARCHITECTURE ON MNIST DATASET

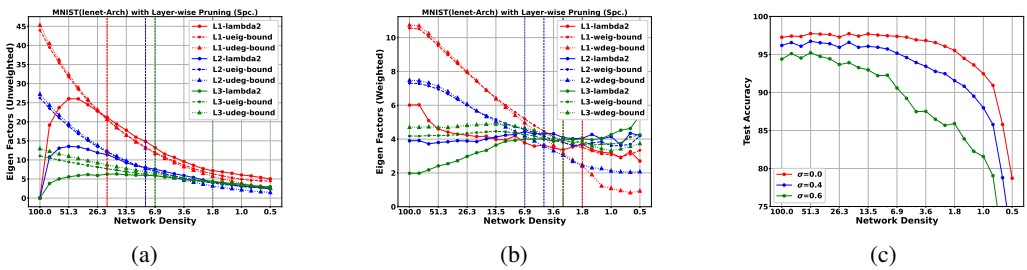

Figure 6: The results for Lenet using MNIST dataset with pruning percentile $p_{fc} = 0.2$ and $p_{out} = 0.1$ (layer-wise-pruning); (a) eigen factors for the unweighted graph, (b) eigen factors for the weighted graph, (c) test accuracy

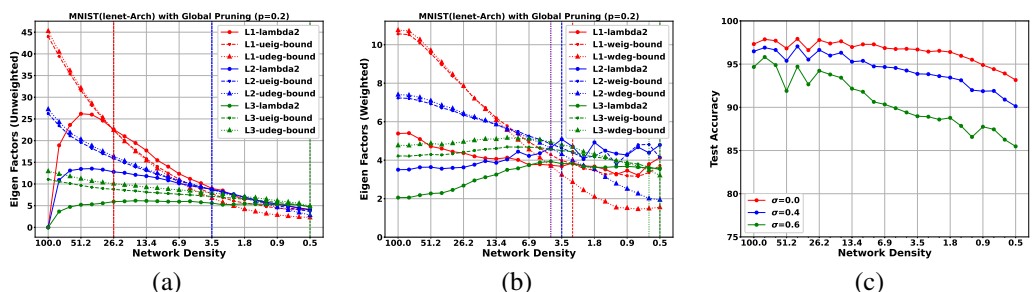

Figure 7: The eigen factors for global pruning with pruning percentile $p = 0.2$ on MNIST dataset using Lenet architecture

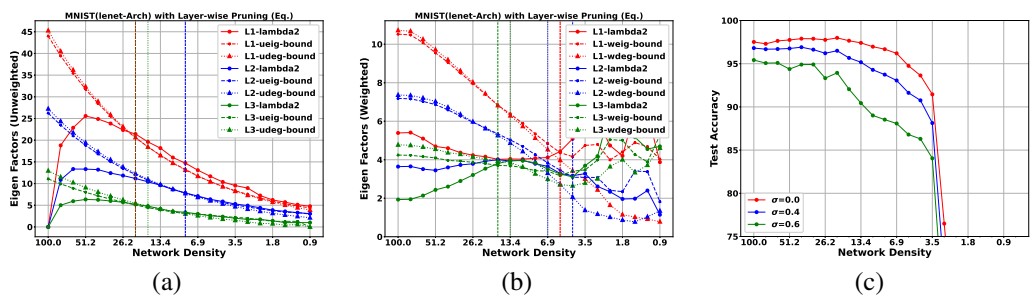

Figure 8: The eigen factors for layer-wise pruning with pruning percentile $p = 0.2$ on MNIST dataset using Lenet architecture

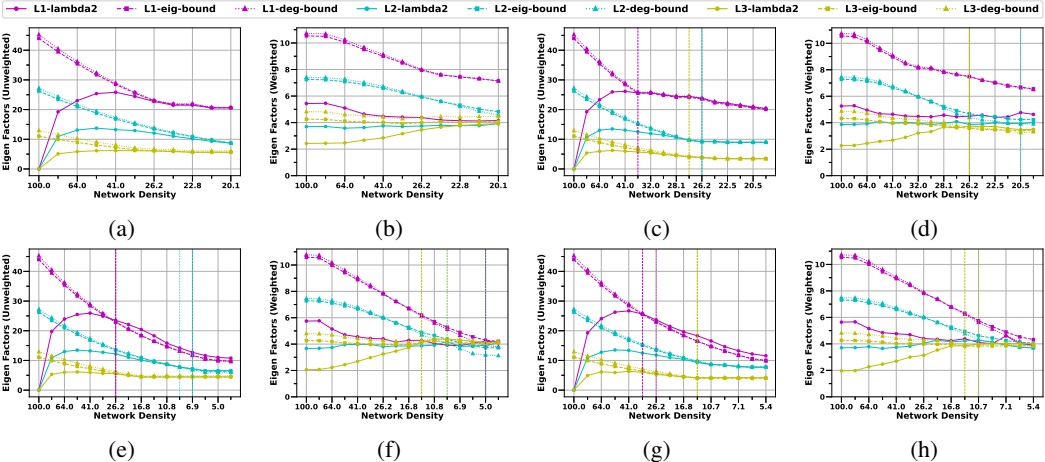

Figure 9: The eigen-factor results for the proposed pruning algorithm on MNIST dataset using Lenet architecture and different types of bound criteria, mentioned in Table 1; (a-b) for **UE**, (c-d) for **UD**, (e-f) for **WE**, (g-h) for **WD** (with average degree > 1)

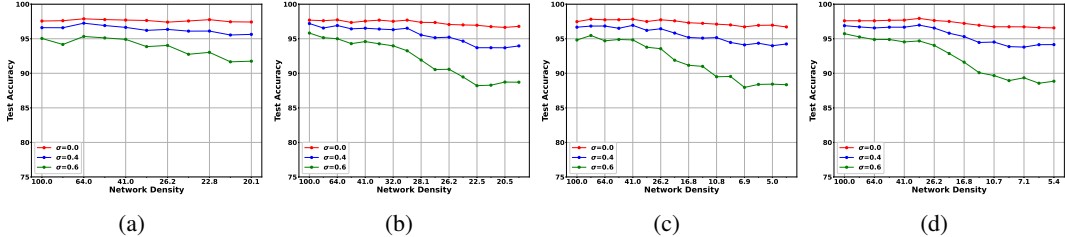

Figure 10: The test accuracy results of the proposed pruning algorithm on MNIST dataset using Lenet architecture with different noise levels in the input image using and bound criteria, mentioned in Table 1; (a) for **UE**, (b) for **UD**, (c) for **WE**, (d) for **WD** (with average degree > 1)

## A.2 MORE RESULTS FOR CONV2 ARCHITECTURE ON CIFAR10 DATASET

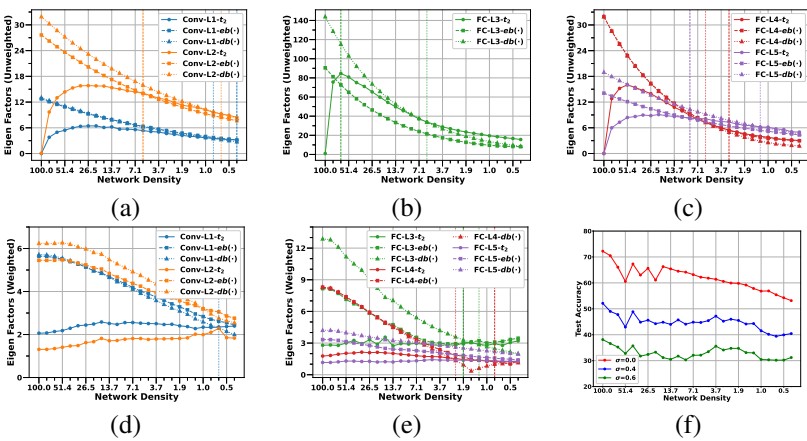

Figure 11: The results for Conv2 architecture using CIFAR10 dataset with pruning percentile $p_{conv} = 0.1$, $p_{fc} = 0.2$, and $p_{out} = 0.1$ (layer-wise-pruning); (a-c) eigen factors for the unweighted graph, (d-e) eigen factors for the weighted graph, (f) test accuracy

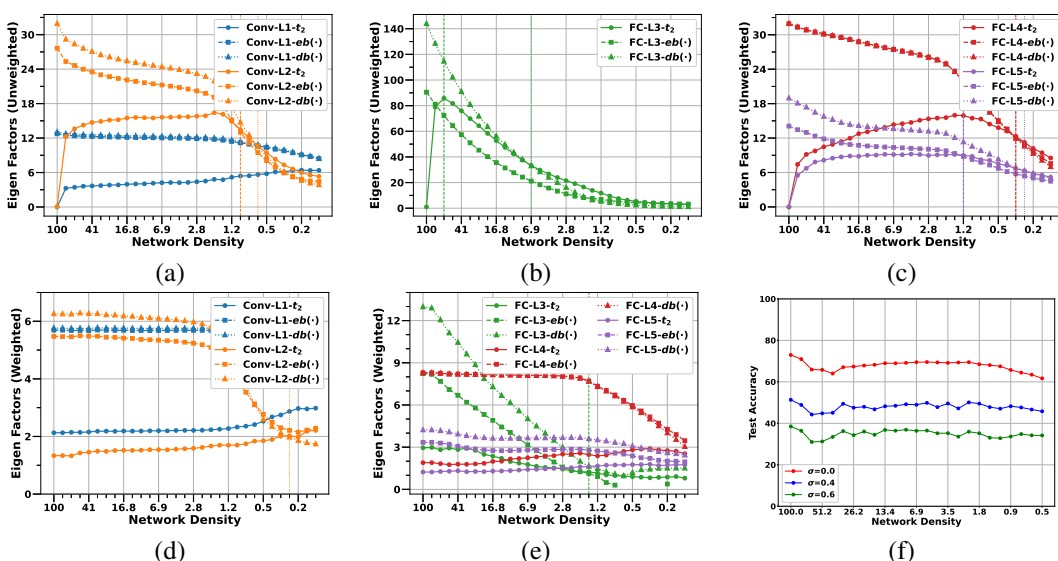

Figure 12: The eigen factors for the IMP pruning algorithm on CIFAR10 dataset using Conv2 architecture; (a-e) Global Pruning with $p = 0.2$

### A.3 MORE RESULTS FOR CONV4 ARCHITECTURE ON CIFAR10 DATASET

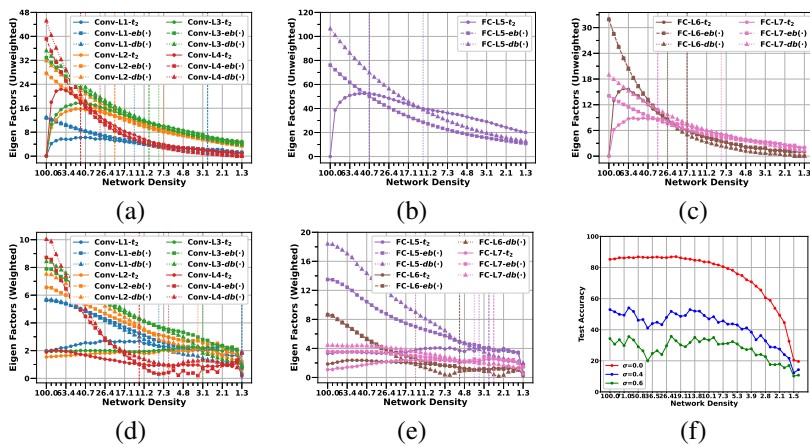

Figure 13: The results for Conv4 architecture using CIFAR10 dataset with pruning percentile $p_{conv} = 0.1$, $p_{fc} = 0.2$, and $p_{out} = 0.1$ (layer-wise-pruning); (a-c) eigen factors for the unweighted graph, (d-e) eigen factors for the weighted graph, (f) test accuracy

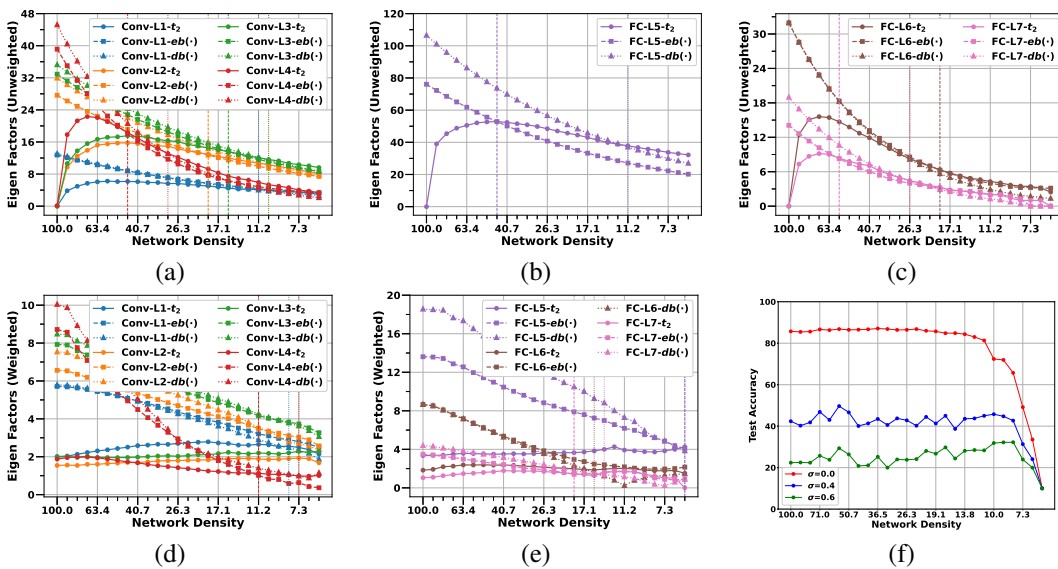

Figure 14: The eigen factors for the IMP pruning algorithm on CIFAR10 dataset using Conv4 architecture; (a-e) Layer-wise Pruning with $p_{conv} = 0.1$, $p_{fc} = 0.2$, and $p_{out} = 0.2$

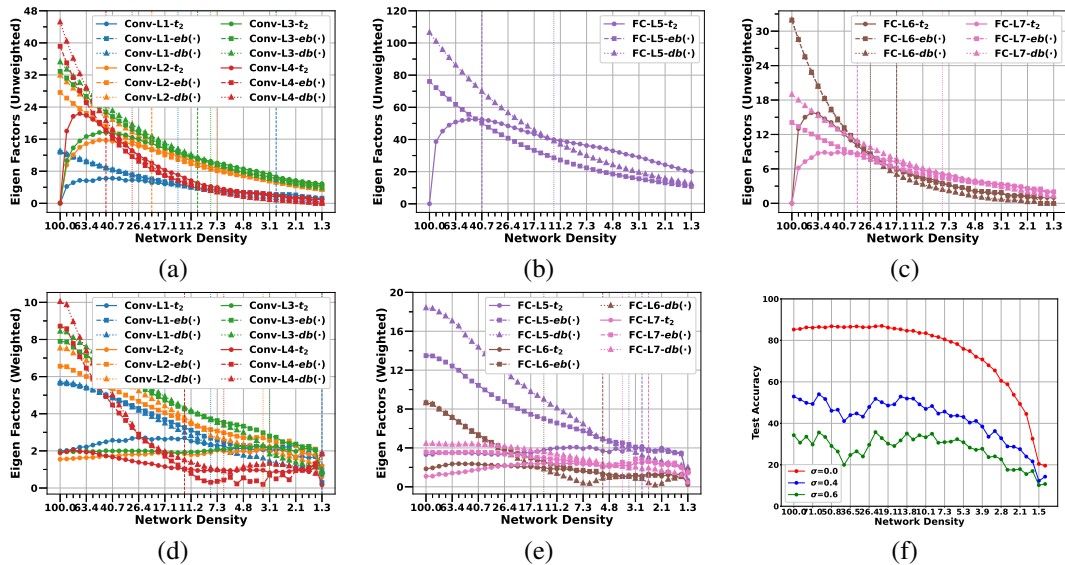

Figure 15: The eigen factors for the IMP pruning algorithm on CIFAR10 dataset using Conv4 architecture; (a-e) Layer-wise Pruning with $p_{conv} = 0.1$, $p_{fc} = 0.2$, and $p_{out} = 0.1$

## A.4 MORE RESULTS FOR CONV6 ARCHITECTURE ON CIFAR10 DATASET

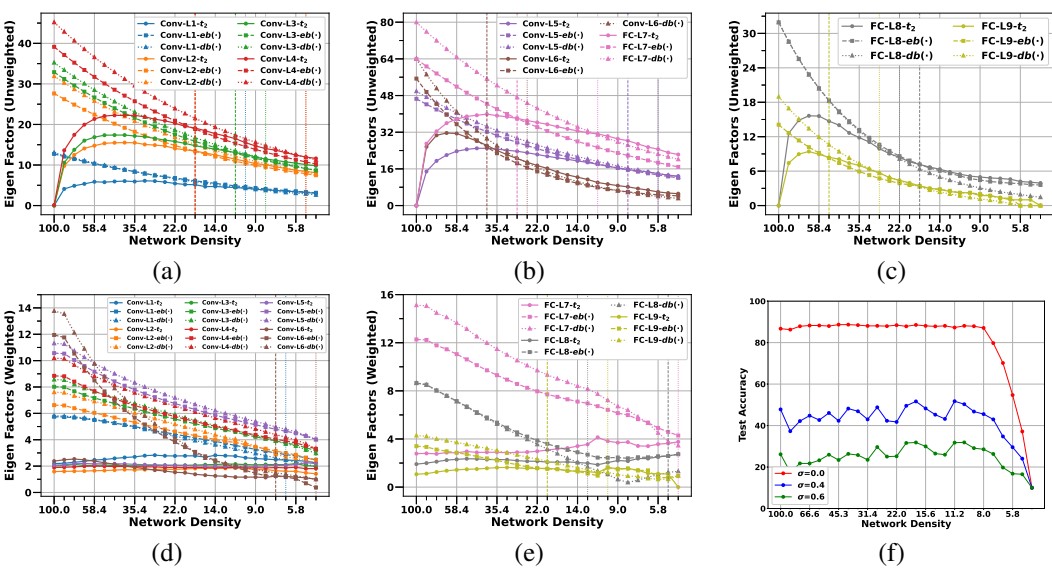

Figure 16: The eigen factors for the IMP pruning algorithm on CIFAR10 dataset using Conv6 architecture; (a-e) Layer-wise Pruning with $p_{conv} = 0.1$, $p_{fc} = 0.2$, and $p_{out} = 0.2$

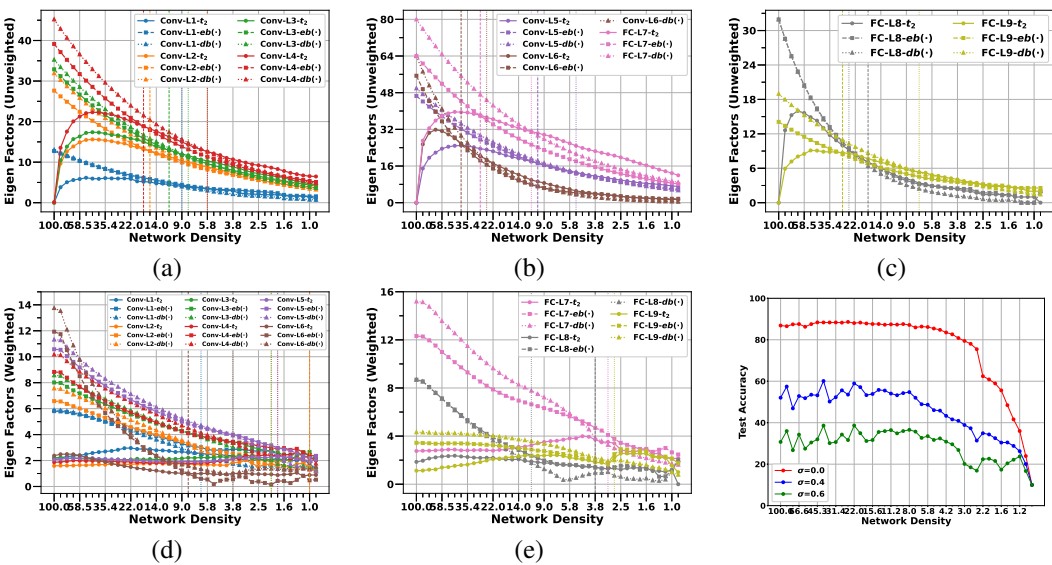

Figure 17: The eigen factors for the IMP pruning algorithm on CIFAR10 dataset using Conv6 architecture; (a-e) Layer-wise Pruning with $p_{conv} = 0.1$, $p_{fc} = 0.2$, and $p_{out} = 0.1$

## A.5 RESULTS FOR VGG19

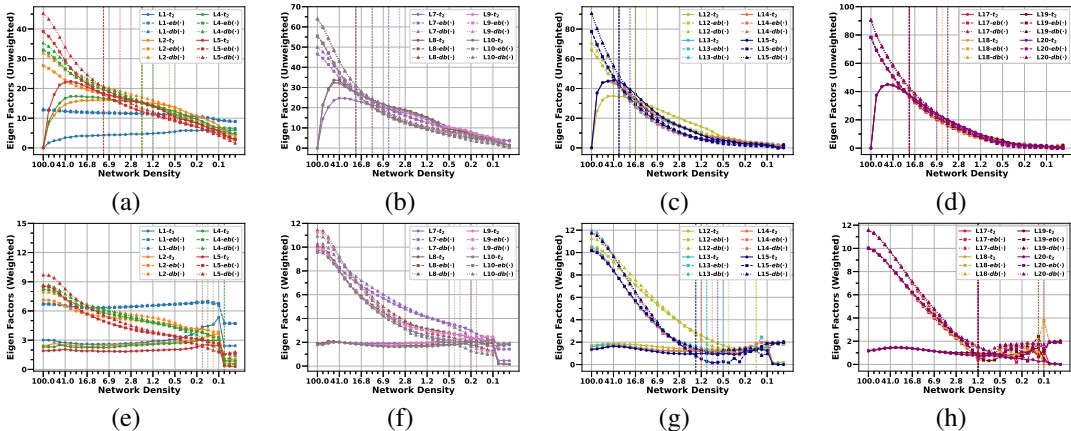

Figure 18: The eigen factors for the IMP pruning algorithm on CIFAR10 dataset using VGG19 architecture and Global Pruning with $p = 0.2$ and $lr = 0.01$

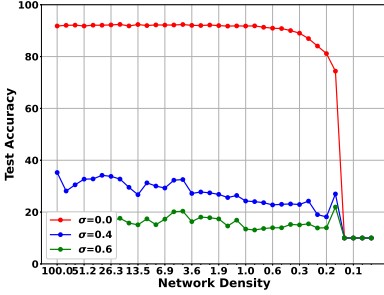

Figure 19: Results for the test accuracy of VGG19 on different noise level

## A.6 HYPER-PARAMETERS DESCRIPTION

Table 3: Hyper-parameter settings for experimenting LTH using iterative magnitude based pruning

|  | Lenet (on MNIST) | Conv4 (on CIFAR10) |
|---|---|---|
| Optimizer | Adam | Adam |
| Training Iterations | 20000 | 25000 |
| Batch size | 60 | 60 |
| Learning Rate | 0.0012 | 0.0003 |
| Pruning epochs | 50 | 50 |
| Model initialization | Kaiming Normal | Kaiming Normal |
| Conv Layers |  | 64,64, pool 128, 128, pool |
| FC layers | 300, 100, 10 | 256, 256, 10 |
| pruning epochs(in comparison) | 60 | 60 |

## A.7 DIFFERENT PARAMETERS VALUES WITH RESPECT TO THE RAMANUJAN GRAPH BASED PRUNING ALGORITHM

Table 4: Representative result of the parameters found in the last pruning epoch of SynFlow-Bound Algorithm on Lenet/MNIST

| | | | |
|---|---|---|---|
| L1 | unweighted | $(t_1, t_2, d_{avg}, d_{avgR}, eb, db, \Delta_S, \Delta_R)$ | (31.84, 14.32, 235.20, 3.07, 11.11, 16.74, -0.22, 0.17) |
| L1 | weighted | $(t_1, t_2, d_{avgL}, d_{avgR}, eb, db, \Delta_S)$ | (19.59, 1.37, 31.63, 12.10, 8.62, 8.87, 5.29) |
| L1 | Score | $(t_1, t_2, d_{avgL}, d_{avgR}, eb, db, \Delta_S)$ | (14.07, 6.96, 99.42, 1.30, 7.23, 10.47, 0.04 ) |
| L1 | | (layer-wise density, remaining/total parameters) | (0.01, 2352 / 235200) |
| L2 | unweighted | $(t_1, t_2, d_{avg}, d_{avgR}, eb, db, \Delta_S, \Delta_R)$ | (12.41, 5.75, 3.37, 30.00, 6.76, 6.92, 0.17, 0.20) |
| L2 | weighted | $(t_1, t_2, d_{avgL}, d_{avgR}, eb, db, \Delta_S)$ | (11.40, 1.34, 19.66, 6.55, 6.45, 6.68, 3.82) |
| L2 | Score | $(t_1, t_2, d_{avgL}, d_{avgR}, eb, db, \Delta_S)$ | (43.27, 21.31, 11.23, 99.93, 13.00, 13.14, -0.39 ) |
| L2 | | (layer-wise density, remaining/total parameters) | ( 0.01, 300.0 / 30000) |
| L3 | unweighted | $(t_1, t_2, d_{avg}, d_{avgR}, eb, db, \Delta_S, \Delta_R)$ | (29.83, 0.00, 89.00, 10.00, 10.74, 12.38, 10356908.93, 11940047.67) |
| L3 | weighted | $(t_1, t_2, d_{avgL}, d_{avgR}, eb, db, \Delta_S)$ | (3.54, 1.01, 10.90, 1.09, 3.19, 3.45, 2.16) |
| L3 | weighted | $(t_1, t_2, d_{avgL}, d_{avgR}, eb, db, \Delta_S)$ | 12.74, 7.83, 9.72, 7.29, 6.85, 5.46, -0.12 |
| L3 | | (layer-wise density, remaining/total parameters) | (0.89, 890 / 1000) |

