# OpenReview forum: "Revisiting the Lottery Ticket Hypothesis: A Ramanujan Graph Perspective"
_ICLR.cc/2022/Conference — ICLR 2022 Submitted_

### Official Review · Reviewer_M3ko · 2021-11-01

**Correctness:** 3
**Technical Novelty And Significance:** 2
**Empirical Novelty And Significance:** 2
**Recommendation:** 5
**Confidence:** 5

**Details Of Ethics Concerns:**

None.

**Main Review:**

**Pros**: The proposed pruning method is based on the spectral graph theory, which is novel and interesting.

**Cons**:
1. **Significance of the method**.
    - What is the practical significance of the proposed pruning method in this work?
    - Whether the proposed method can effectively improve the pruning performance, such as alleviating the performance degradation?
    - Whether the winning lottery ticket can be found more effectively?
    -  For pruning the neural network, what is the significance of robustness against the noise in the dataset? Maybe we should focus on pruning instead of robustness.

2. **More comparisons are needed**.
    -  This paper does not show the performance of other state-of-the-art pruning methods, such as IMP or gradient-based pruning.
    -  How robust are other pruning methods to noise?
    -  Authors should also make more ablation studies to demonstrate the roles or impact of different components of the proposed method.

3. **Other issues**.
    -  The experiment section in this paper is too single. The results shown in Figures 3, 4, and 5 are all similar, except that different datasets and models are changed.  This section should include more important results to demonstrate the proposed arguments. Please put the insignificant results or curves to the appendix.
    -  The experiment section frequently shows the curves of eigen factors. What does this indicate? What kind of enlightenment does this changing trend provide us?
    -  Why does the proposed method establish robustness against the noise?

**Summary Of The Paper:**

In this paper, the authors study the properties of the subnetworks in lottery ticket hypothesis (LTH) from the perspective of the spectral graph theory.
They argue that the pruned network in LTH remains a Ramanujan graph.
The performance of the subnetworks begins to degrade with the loss of Ramanujan graph property.
To this end, they propose an eigen-bound based network pruning algorithm to preserve the graph property.
Extensive experiments are conducted to demonstrate the effectiveness of the proposed method.

**Summary Of The Review:**

In general, the paper is well written. I hope the authors should first answer the aforementioned questions cautiously and carefully.

---

> ### Author Response · Authors · 2021-11-22
> **Response to Reviewer**
>
> We thank the reviewers for the valuable comments.
>
> Comment 1: What is the practical significance of the proposed pruning method in this work?
> Whether the proposed method can effectively improve the pruning performance, such as alleviating the performance degradation? Whether the winning lottery ticket can be found more effectively.
>
> Response: We do not propose a new pruning algorithm rather we demonstrate that the Ramanujan graph properties may be considered by pruning algorithms for better preservation of test accuracy for sparse networks. In the modified iterative pruning algorithm presented by us the Ramanujan graph criteria helps identify layers that are still amenable to further pruning. The method prunes such layers while avoiding pruning in layers that have already lost their Ramanujan property. This helps achieve a higher global sparsity of the pruned networks while maintaining the test accuracy.
>
> The effect of this criteria on the popular IMP, SNIP and SynFlow pruning algorithms are shown in Table 2.
>
> Comment 2: For pruning the neural network, what is the significance of robustness against the noise in the dataset? Maybe we should focus on pruning instead of robustness.
>
> Response: We agree on this point. We have presented a comparison of pruning performance in Table 2.
>
> Comment 3: More comparisons are needed. This paper does not show the performance of other state-of-the-art pruning methods, such as IMP or gradient-based pruning.
>
> Response: We present comparison with state-of-art pruning techniques in Table 2. The algorithms compared are the popular IMP, SNIP,  and SynFlow algorithms. It may be noted that the proposed Ramanujan graph criteria may be used along with any existing pruning algorithm .
>
> Comment 4: How robust are other pruning methods to noise?
> Response: Most of the existing pruning methods do preserve Ramanujan graph properties for moderate network densities. We expect noise robustness for them too.
>
> Comment 5: Authors should also make more ablation studies to demonstrate the roles or impact of different components of the proposed method
>
> Response: We have augmented Figures 4 and 5 to include results pertaining to various components.
>
> Comment 6: The experiment section in this paper is too single. The results shown in Figures 3, 4, and 5 are all similar, except that different datasets and models are changed. This section should include more important results to demonstrate the proposed arguments. Please put the insignificant results or curves to the appendix.
>
> Response: We have redrawn and combined a number of figures. Please see Figures 4 and 5 which demonstrate the main claims. The remaining results are now moved to the Appendix.
>
> Comment 7: The experiment section frequently shows the curves of eigen factors. What does this indicate? What kind of enlightenment does this changing trend provide us?
> Response: We have redrawn the figures using a more succinct form of the eigen factors, namely the bound gaps. The main claim about the Ramanujan graph properties of the winning lottery tickets is now supported by Figures 4 and 5.
> We have to tried to show the parameters related to the Ramanujan graph property in each layer. Which is now presented in Appendix.
>
> Comment 8: Why does the proposed method establish robustness against the noise?
> Response: Ramanujan graphs are resilient and sparse networks that preserve connectivity. As is illustrated in the small example in Figure 3, preservation of Ramanujan graph properties for all the network layers leads to the resilience of information flow paths across the network even for sparse pruned networks. In literature it has been observed that preservation of such flow paths lead to improved noise robustness [Liu 2018].

---

### Official Review · Reviewer_u9fu · 2021-11-01

**Correctness:** 3
**Technical Novelty And Significance:** 2
**Empirical Novelty And Significance:** 2
**Recommendation:** 5
**Confidence:** 3

**Main Review:**

While the connection of Ramanujan expander graphs to lottery tickets is interesting, the experiments are not convincing and leave doubt that this is even a relevant property for network pruning. The practical relevance of the stopping criterion is doubtfull. Detailed points of critique follow below.

Strengths:
+ The idea to measure the potential of a pruned network to serve as lottery ticket based on the eigenspectrum is novel and interesting.
+ The idea that the Ramanujan graph property could be beneficial for sparse network structures could be also useful (but this remains open to show).

Weaknesses:
- The introduction into Ramanujan graphs is lengthy and only motivational, as the results on maximality hold only for regular graphs and not for the lottery tickets (which are usually irregular).
- The authors claim that they introduce a new pruning algorithm, while in fact they only propose a stopping criterion for iterative magnitude pruning (IMP), which requires the repeated computation of the two largest eigenvalues of potentially large weight matrices. The original IMP is computationally expensive already, which leaves the question what the added value of the eigen-bound based stopping criterion should be?
- As a stopping criterion for pruning, it is not relevant, as the validation set performance (or even training set performance) is in fact more indicative.
- Is the contribution should be the insight that graphs with the Ramanujan property are better suitable as lottery tickets than other structures, I would expect different experiments that test this hypothesis.
For instance, I would expect that random graphs are sampled or otherwise constructed that have comparable properties (like density, degree distribution, flow preserving, etc.)
which either have the Ramanujan graph property or not (but are otherwise comparable). Or through rewiring of edges the Ramanujan graph property can be gained or lost...
For both these sets (Ramanujan or not) one could then test whether they are suitable lottery tickets (or at least preserve information flow).
- For real practical impact, I would then expect an algorithm that repairs a pruned network to gain back the Ramanujan graph property (by edge rewiring that preserved the overall network density) and then would want to see experiments that suggest a superior performance of this pruning algorithm with repair.
- Different from the made claims, the Figures 3, 4, 5 do not even seem to indicate a strong link between the loss of the Ramanujan graph property and a loss of accuracy.
I would expect to see a sharp phase transition and complete loss of network accuracy at the point where the Ramanujan graph property is lost if the made claims were true.
- Or is simply the connectivity of the graph the relevant property?
I would expect at least a comparison with methods like Synflow or layerwise pruning approaches that try to prevent layer collapse and preserve information flow through the network or orthogonal repair as in "A Signal Propagation Perspective for Pruning Neural Networks at Initialization", ICLR 2020.

Points of minor critique and open questions:
- Definition 3 for irregular graphs: Two replacements of the same quantity (the regularity r) are proposed, which is contradictory in this form. This could be explained more precisely.
- Multiple symbols, abbreviations, figure legends, etc. are nowhere defined. For instance, \sigma shall probably refer to the standard deviation of the applied Gaussian noise?
- Figure 7: It is impossible to compare which pruning algorithms perform best. They all look very similar.
- No error bars are reported anywhere.


**Summary Of The Paper:**

The authors state the hypothesis that the Ramanujan graph property is necessary or at least beneficial for pruned networks to serve as lottery tickets. Motivated by this claim, they propose a stopping criterion for iterative magnitude pruning, which is based on the two largest eigenvalues of the weight (or adjacency) matrices of neural networks and should (approximately) preserve the Ramanujan graph property.


**Summary Of The Review:**

As the merit of the proposed eigen-bound based pruning is unclear, I believe the work is not suitable for publication at ICLR in this form.

---

> ### Author Response · Authors · 2021-11-22
> **Response to Reviewer - Part I**
>
> We thank the reviewer for the for the insightful comments that helped us to improve the manuscripts.
>
> Comment 1: While the connection of Ramanujan expander graphs to lottery tickets is interesting, the experiments are not convincing and leave doubt that this is even a relevant property for network pruning.
>
> Response: We have updated the experimental results to support our claims.
>
> We have redrawn the figures so that the experimental verification of the connection between Ramanujan graphs and the LTH is observable. Please see Figures 4 and 5.
>
> The relevance of the properties from the perspective of pruning is demonstrated in Table 2. We observe that three popular pruning algorithms IMP, SNIP, and SynFlow maintain the Ramanujan graph property till the point when accuracy begins to drop sharply.
>
>
> Comment 2: The practical relevance of the stopping criterion is doubtfull.
>
> Response: We have shown that the Ramanujan Graph property preservation criteria can augment the network pruning algorithms. Please refer to Table 2. We observe that the violation of spectral bounds does indicate a significant loss of test set accuracy. We identified networks obtained by IMP pruning that are non-Ramanujan vis-a-vis those that are obtained by SynFlow-Bounded and are Ramanujan. Even though both these networks have almost the same sparsity, the accuracy of the non-Ramanujan networks is much less.  The plots for LTH in Figure 4 and 5 justify the claim.
>
> Comment 3: The introduction into Ramanujan graphs is lengthy and only motivational, as the results on maximality hold only for regular graphs and not for the lottery tickets (which are usually irregular).
>
> Response: Generalization of Ramanujan graphs to irregular graphs using the notion of universal cover is described. Details are provided in the reference Hoory et al, 2006.
>
> Comment 4: The authors claim that they introduce a new pruning algorithm, while in fact they only propose a stopping criterion for iterative magnitude pruning (IMP), which requires the repeated computation of the two largest eigenvalues of potentially large weight matrices. The original IMP is computationally expensive already, which leaves the question what the added value of the eigen-bound based stopping criterion should be?
>
> Response: We propose a Ramanujan property preserving pruning algorithm that can be used with other weight score functions suggested by the pruning algorithms like SNIP, SynFlow etc. The algorithm identifies layers that are amenable to pruning while avoiding pruning in layers that have lost their Ramanujan property. Figures 4 and 5 show that loss of Ramanujan property degrades the performance of pruned networks with a similar percentile of remaining weights.
>
> Verifying the spectral criteria involves computing the eigenvalues and average degree for the bi-adjacency matrices of the layers. In our bi-partite graph formulation, the adjacency matrices are the same as the size of the kernels for the convolution layers (Figure 3) and the number of nodes for the fully-connected layers. Since the kernel size as well as the number of fully connected layers are small for most networks the overhead is not much. However, the drawback exists for wide-layers.
>
> Comment 5: As a stopping criterion for pruning, it is not relevant, as the validation set performance (or even training set performance) is in fact more indicative.
>
> Response: We argue that the Ramanujan graph preservation criteria leads to networks that are more robust to noise as compared to those obtained using the validation accuracy criteria. The correlation of noise robustness with the spectral criteria is shown in Figures 4 and 5.

---

> > ### Author Response · Authors · 2021-11-22
> > **Response to Reviewer - Part II**
> >
> > Comment 6: Is the contribution should be the insight that graphs with the Ramanujan property are better suitable as lottery tickets than other structures, I would expect different experiments that test this hypothesis. For instance, I would expect that random graphs are sampled or otherwise constructed that have comparable properties (like density, degree distribution, flow preserving, etc.) which either have the Ramanujan graph property or not (but are otherwise comparable). Or through rewiring of edges the Ramanujan graph property can be gained or lost... For both these sets (Ramanujan or not) one could then test whether they are suitable lottery tickets (or at least preserve
> > information flow).
> >
> > Response: The primary contribution of the paper is the empirical demonstration that winning tickets often satisfy the Ramanujan graph property. This is shown in Figures 4 and 5 where we demarcate variation of test set accuracy with network density into three regions based on validity of spectral bounds. In the first region all the bipartite layers of the network are Ramanujan graphs and there is a little loss of accuracy. In the second region some of the layers have lost the property, loss of accuracy is slightly more for clean data and significant for noisy data. In the third region all the layers lose the Ramanujan property and accuracy drops sharply. Based on these observations we propose a modified iterative pruning algorithm which attempts to preserve Ramanujan property for as many layers as possible by using the spectral bound criteria to determine the layers which are still amenable to pruning. The approach may as well be used with the magnitude of weight score functions suggested by other pruning algorithms like SynFlow and not just the weight magnitudes.
> >
> >  We identified networks obtained by IMP pruning that are non-Ramanujan vis-a-vis those that are Ramanujan. Even though both these networks have the same sparsity, the accuracy of the non-Ramanujan networks is much less. A similar behaviour is expected for random graphs with high sparsity.
> >
> > Comment 7: For real practical impact, I would then expect an algorithm that repairs a pruned network to gain back the Ramanujan graph property (by edge rewiring that preserved the overall network density) and then  would want to see experiments that suggest a superior  performance of this pruning algorithm with repair
> >
> > Response: Repairing the network would be our next goal. We would like to point out that construction of Ramanujan graphs is still a non-trivial problem. We believe that network architecture search might help us repair the pruned networks. We would like to consider this in our future study.
> >
> > Comment 8: Different from the made claims, the Figures 3, 4, 5 do not even seem to indicate a strong link between the loss of the Ramanujan graph property and a loss of accuracy. I would expect to see a sharp phase transition and complete loss of network accuracy at the point where
> > the Ramanujan graph property is lost if the made claims were true. Or is simply the connectivity of the graph the relevant property?
> >
> > Response: The significance of Ramanujan graph property of the network layers is explained in the small example of Figure 3.
> >
> > Figures in the experimental results section are now redrawn to remove the confusions. Please see Figures 4 and 5. We agree that a sharp phase change is not observed. However, the transition points of the Ramanujan graph properties do indicate the beginning of loss of accuracy.

---

> > > ### Author Response · Authors · 2021-11-22
> > > **Response to Reviewer - Part III**
> > >
> > > Comment 9: Would expect at least a comparison with methods like Synflow or layerwise pruning approaches that try to prevent layer collapse and preserve information flow through the network or orthogonal repair as in "A Signal Propagation Perspective for Pruning Neural Networks at
> > > Initialization", ICLR 2020."
> > >
> > > Response: We have presented a comparison with the Synflow algorithm. Please refer to Table 2. It may be noted that the Ramanujan graph properties are preserved in the pruned networks obtained by the Synflow algorithm, while they are not preserved for random pruning. As we have mentioned in the paper, we don’t present an entirely new pruning algorithm, we suggest that the Ramanujan graph properties may guide the pruning algorithms. The property helps us identify layers that are still amenable to pruning while preserving the Ramanujan graph property.
> > >
> > >
> > > Comment 10: Definition 3 for irregular graphs: Two replacements of the same quantity (the regularity r) are proposed, which is contradictory in this form. This could be explained more precisely.
> > >
> > > Response: The notion of unweighted irregular Ramanujan graphs is related to the notion of universal cover. For unweighted graphs, large expansion property is guaranteed by either considering the average degree bound (follows by considering the universal cover) or the eigenvalue bound (by the discrete Cheeger inequality). For weighted graphs, the eigenvalue bound ensures large expansion. These have now been clarified in the section, along with the appropriate references.
> > >
> > > Comment 11: Multiple symbols, abbreviations, figure legends, etc. are nowhere defined. For instance, \sigma shall probably refer to the standard deviation of the applied Gaussian noise
> > >
> > > Response: Apologies for the confusing presentation. We have now redrawn the figures and defined the terms marked in the figures.
> > >
> > >
> > > Comment 12: Figure 7: It is impossible to compare which pruning algorithms perform best. They all look very similar.
> > >
> > > Response: Figure 7 is removed. We have added Tables 2 to explain the efficacy of the pruning algorithms.
> > >
> > > Comment 13: No error bars are reported anywhere.
> > >
> > > Response: We have plotted the error bars in all the figures over 5 runs

---

> > > > ### Comment · Reviewer_u9fu · 2021-11-25
> > > > **I thank the authors for the detailed comments.**
> > > >
> > > > I appreciate the comments by the authors and acknowledge the update of the manuscript.
> > > > While I find the idea and motivation of the paper quite interesting, I still cannot recommend acceptance for publication at this stage, because I cannot see a strong empirical correlation between the loss of the Ramanujan graph property and reduction of lottery ticket accuracy. There are many possible alternative explanations for the decreasing accuracy. However, I see potential of this work if the authors can derive repair algorithms or test in more refined experiments (as described above) whether really the Ramanujan graph property is the key lottery ticket property.

---

> > > > > ### Author Response · Authors · 2021-11-26
> > > > > **Response to Reviewer**
> > > > >
> > > > > As suggested in the initial round of reviews, we had investigated and tested the relation between Ramanujan graph property and test accuracy of networks with similar density (remaining weights). This is reported in the manuscript in page 9, paragraph 3.
> > > > >
> > > > > For better clarity and reinforcing the fact, we provide a table below of accuracy, network density, and the number of layers that have lost the Ramanujan graph property for two CIFAR10/Conv4 networks pruned by IMP and SynFlow algorithms. It is clearly seen that SynFlow retains the Ramanujan graph property, and hence, the accuracy for a wide range of network density [10\%-1\%], while IMP loses the Ramanujan property negatively impacting the accuracy.  Between the two networks having the same density, the one better preserving the Ramanujan property has a higher accuracy.
> > > > >
> > > > > $$ \ \ \ \ \ \ \ \ \ \ \ \ \ \ \ \ \ \  \ \ \ \ \ \ \ \ \ \ \ \ \ \ \ \ \ \ \ \ \ \ \ \  \ \ \ \ \ \  \ \ \ \ \ \ \ \ \ \ \ \ \ \ \ \ \ \  \ \ \ \ \ \ \text{Table} $$
> > > > > $$ Density \ \ \ \ IMP(Accuracy) \ \ \ \ IMP(X) \ \ \ \ SynFlow(Accuracy) \ \ \ \ SynFlow (X)$$
> > > > > 	$$ 10.00 \ \ \ \  \ \ \ \ \ \ \ \ \ \ 81.91 \ \ \ \ \ \ \ \ \ \ \ \ \ \ \ \ \ \  \ \ \ \ \ \ 1 \ \ \ \ \ \ \ \ \ \ \ \ \ \ \ \ \ \ \ \ \ \ \ \ \  85.10 \ \ \ \  \ \ \ \ \ \ \ \ \ \ \ \ \ \ \ \ \ \ \ \ 0 $$
> > > > > 	$$ 03.16 \ \ \ \  \ \ \ \ \ \ \ \ \ \ 60.61 \ \ \ \ \ \ \ \ \ \ \ \ \ \ \ \ \ \  \ \ \ \ \ \ 2 \ \ \ \ \ \ \ \ \ \ \ \ \ \ \ \ \ \ \ \ \ \ \ \ \  77.18  \ \ \ \  \ \ \ \ \ \ \ \ \ \ \ \ \ \ \ \ \ \ \ \ 1 $$
> > > > > 	$$ 01.00 \ \ \ \  \ \ \ \ \ \ \ \ \ \ 10.00 \ \ \ \ \ \ \ \ \ \ \ \ \ \ \ \ \ \  \ \ \ \ \ \ 4  \ \ \ \ \ \ \ \ \ \ \ \ \ \ \ \ \ \ \ \ \ \ \ \ \   69.20  \ \ \ \  \ \ \ \ \ \ \ \ \ \ \ \ \ \ \ \ \ \ \ \ 3  $$
> > > > > $$ X =  \text{Number of layers that have lost the Ramanujan graph property} $$

---

> > > > > > ### Comment · Reviewer_u9fu · 2021-11-29
> > > > > > **Not enough evidence for accept**
> > > > > >
> > > > > > I appreciate that the authors highlighted this additional table and I increased my score accordingly.
> > > > > > However, these results do not convince me, as the the synflow algorithm could simply be superior to IMP in this context for different reasons. Also the reduction of accuracy with decreasing density is not surprising.

---

> > > > > > > ### Author Response · Authors · 2021-11-29
> > > > > > > **Response to Reviewer**
> > > > > > >
> > > > > > > We thank the reviewer for updating the score. It is well known in the literature that SynFlow is better than IMP in terms of test accuracy. Our results highlight that it also performs better in maintaining the quality of connectivity following the Ramanujan Graph property. Consideration of the Ramanujan graph property along with the SynFlow algorithm further leads to improvement of accuracy as can be seen from the results in Table 2 of the paper.

---

> > > > > > > > ### Comment · Reviewer_u9fu · 2021-11-30
> > > > > > > > **Response to authors**
> > > > > > > >
> > > > > > > > I acknowledge this insight and appreciate the discussion.
> > > > > > > > To clarify my argument: SynFlow has also been designed to preserve the flow and the overall connectivity of the network. Maybe that is already enough to explain why it performs better.

---

### Official Review · Reviewer_rGMS · 2021-11-02

**Correctness:** 3
**Technical Novelty And Significance:** 2
**Empirical Novelty And Significance:** 2
**Recommendation:** 5
**Confidence:** 3

**Main Review:**


Strengths:
- The paper builds upon a nice premise by introducing expander graphs to describe connectivity between neural network layers.
- Choosing the pruning termination criterium based on on the extended spectral gap (Algorithm 1) is a good idea and novel to my knowledge.

Weaknesses:
- The experiments are not able to convince me that the chosen bound leads to good pruning results. In some experiments they seem to fit, in others they do not.
- There is also no solid theoretical justification, why the termination criterium should fit (except for the already known correlation between the spectral gap and density).
- The work is lacking comparisons to previous pruning methods, showcasing the usefulness and significance of the proposed algorithm.
- The experiment section is confusing (see below)

Issues in Experiments section:
- I have trouble following the figures. The labels in the figures are not consistent with the names in the text (lambda vs t, L1-L3 not defined). Minor: the figure labels are way too small.
- UE, UD, WE, WD only defined in a Table and the text is lacking clear discussions and conclusions for the results in the different scenarios.
- It would help alot if the test accuracy figures would show the point in which Algorithm 1 would stop the pruning.
- If I understood correctly, it should be easy to apply Algorithm 1 to the networks and present the resulting test accuracy in comparison to previous pruning techniques. However, that is not done.


**Summary Of The Paper:**

The work proposes to utilize the spectral gap of the bipartite network connectivity graphs between neural network layers to terminate an iterative weight pruning algorithm.

**Summary Of The Review:**

All in all, while I like the general idea of utilizing spectral graph information of bipartite network connectivity graphs for pruning decisions, I find the paper is lacking a clear message and the required experimental evidence to back it up. There are no comparisons to previous pruning techniques, which would allow to showcase significance. Therefore, I tend to vote for rejection.

---

> ### Author Response · Authors · 2021-11-22
> **Response to Reviewer**
>
> We thank the reviewer for the acknowledging the novelty of the contribution and identifying the strength of the work. We address each of the weaknesses mentioned.
>
>
> Comment 1: The experiments are not able to convince me that the chosen bound leads to good pruning results. In some experiments they seem to fit, in others they do not
>
> Response: We have reorganized the plots so that the claimed properties can be observed. See Figures 4 and 5. We notice a fair degree of agreement with the experimental results.  Apologies for the confusing plots in the previous version. Also, it may be noted that we do not present a new pruning algorithm altogether. We claim that existing pruning algorithms like IMP/SynFlow may benefit from using the Ramanujan graph criteria.
>
> Comment 2: There is also no solid theoretical justification, why the termination criteria should fit (except for the already known correlation between
> the spectral gap and density).
>
> Response: We agree that our study is purely empirical. We have presented a small illustrative example to justify our claims in Figure 2. and shown in experimental results with new plots in Figure 4 and 5
>
> Comment 3: The work is lacking comparisons to previous pruning methods, showcasing the usefulness and significance of the proposed algorithm.
>
> Response: We have presented a comparison with three existing pruning algorithms in Tables 2. We agree that we do not present an altogether new pruning algorithm, rather we augment existing pruning algorithms using the Ramanujan graph properties.
>
> Comment 4: Issue in readability of the label size in the plots.
>
> Response: We have reorganized and redrawn the figures. Please refer to Figures 4 and 5 in the revised manuscript.
>
> Comment 5: UE, UD, WE, WD only defined in a Table and the text is lacking clear discussions and conclusions for the results in the different scenarios
> Response: We have removed those terms. The text is rewritten to clarify the confusion.
>
> Comment 6: It would help a lot if the test accuracy figures would show the point in which Algorithm 1 would stop the pruning.
>
> Response: We report the accuracy values and corresponding densities for these points in Table 2.
>
>
> Comment 7: If I understood correctly, it should be easy to apply Algorithm 1 to the networks and present the resulting test accuracy in comparison to previous pruning techniques. However, that is not done.
>
> Response: Comparisons with three other popular pruning algorithms (IMP, SNIP, SynFlow) are now presented in Table 2. Also, it may be noted that we do not present a new pruning algorithm altogether. We claim that existing pruning algorithms like IMP, SNIP, and SynFlow may benefit from using the spectral criteria.
>
> Comment 8: lacking a clear message and the required experimental evidence to back it up. There are no comparisons to previous pruning techniques, which would allow to showcase significance
>
> Response: The primary contribution of the paper is the empirical demonstration that winning tickets often satisfy the Ramanujan graph property. This is shown in Figures 4 and 5 where we demarcate variation of test set accuracy with network density into three regions based on validity of spectral bounds. In the first region all the bipartite layers of the network are Ramanujan graphs and there is a little loss of accuracy. In the second region some of the layers have lost the property, loss of accuracy is slightly more for clean data and significant for noisy data. In the third region all the layers lose the Ramanujan property and accuracy drops sharply. Based on these observations we propose a modification to iterative pruning algorithms which attempts to preserve Ramanujan property for as many layers as possible by using the spectral bound criteria to determine the layers which are still amenable to pruning. The approach may as well be used along with weight score functions suggested by other pruning algorithms.

---

> > ### Comment · Reviewer_rGMS · 2021-11-26
> > **The paper was improved - rating adjusted - significance still limited**
> >
> > I thank the authors for the answers and the improvements made in the paper. The paper is now structured well and the experiments are well-presented.
> > It can be observed that the stopping criteria based on the Ramanujan graph property does lead to a good trade-off between density and test accuracy, which is slightly better than the provided baselines of specific densities. However, I wonder how it compares to simply stopping based on validation accuracy for example, as this would be a very cheap alternative. What I unfortunately still not see is a justification why stopping at the loss of Ramanujan graph property is preferable in comparison to other criteria. In the graphs I do not find a conclusive relation between loss of the Ramanujan graph property and a exact sharp decline of the test accuracy, which leads me to the conclusion that it is only one of many properties that is lost when iteratively pruning the network.
> >
> > All in all, I still have to recommend to reject the paper in the current state, as the impact and insights it provides are not significant enough yet. However, the general idea is interesting and makes sense and I think there might be a more concrete relationship to be found, or more sophisticated algorithms to be designed in future iterations.
> >
> > I increased my score to reflect the improvements made on the paper.

---

> > > ### Author Response · Authors · 2021-11-26
> > > **Response to Reviewer**
> > >
> > > 1. Validation accuracy is a global property of the network. We can use it to stop pruning altogether in the network. Ramanujan graph property pertains to individual layers of the network. It can be used to stop pruning in a particular layer locally, while continuing pruning in
> > > other well/redundantly connected layers. It potentially leads to a lower overall network density while preserving connectivity.
> > >
> > > 2. We observe that a sharp drop in accuracy occurs only in the case of layer collapse where an entire layer is prematurely disconnected. This phenomenon has also been reported in the SynFlow study (Tanaka et al 2020, Figure 1). The pruning algorithms we considered avoid such layer collapse and thus never result in a catastrophic accuracy loss.
> > > We admit that a sharp decline in test accuracy is not observed in our experiments (Figures 4 and 5). In these experiments we have
> > > followed exactly the same iterative pruning strategy IMP as used in LTH study of Frankle and Carbin 2019. A smooth reduction of accuracy is observed in the Frankle and Carbin 2019 study as well. A steady decline in accuracy is observed in our experiments which relates to loss of the Ramanujan graph property.
> > >
> > >
> > > 3. We agree that accuracy may depend on several properties other than Ramanujan graph one. However, this is the single most important
> > > characterization of a network connected to the free flow of information without any bottlenecks as supported by theoretical considerations.
> > >
> > > 4. We thank the reviewer for the valuable suggestions.

---

### Official Review · Reviewer_GGCK · 2021-11-03

**Correctness:** 3
**Technical Novelty And Significance:** 4
**Empirical Novelty And Significance:** 3
**Recommendation:** 6
**Confidence:** 3

**Main Review:**

Strengths:
- as far as I know, the connection from the LTH to expander graphs is both novel and very interesting
- the exposition and algorithm definition are clear and easy to read
- judging from the provided experiments, the analysis seems to be well-supported by data

Weaknesses:
The main weakness is in the experiments section, which is very hard to parse. Problems include:
- very small graphs requiring constant zooming in and out, especially since some curves are only distinguished by line and marker style
- too many curves per graph, some being borderline unreadable (e.g. Figure 5.d)
- the color schemes are confusing: the same colors are used for distinguishing Lenet layers (Figure 3.a, b) and noise levels (Figure 3.c)
- the loss of the ramanujan property should be present on the testing accuracy graphs
- the legend scheme for the curves should be explained more clearly: as of now, they are quite hermetic and require good knowledge of the studied networks' architecture.

Remarks:
- in some cases (e.g. below Figure 1), $p$ switches between the proportion of edges pruned and the proportion of edges kept
- I understand why you wouldn't want a rigorous definition of a universal cover, but you should add a reference
- the theory of Ramanujan graphs is only stated for positive weights, yet this is not the case for neural networks: how is the ramanujan bound defined?
- typo below def. 3: it is the Alon-Boppana theorem

**Summary Of The Paper:**

This paper revisits the Lottery Ticket Hypotheis (LTH), where aggressively pruning a neural network in a smart way allows it to retain its convergence properties, at a lesser computational cost.

The authors advance an explanation relating this LTH to expanders, graphs that have the property that information circulates really well between the vertices, and then to Ramanujan graphs. They then propose an algorithm built on this premise, that prunes the graph as long as the ramanujan property is still satisfied.

Finally, a wealth of experiments is shown, showing the relation between the Ramanujan property and the performance of the pruned network. The authors also test the performance of their Ramanujan pruning algorithm against the same datasets.

**Summary Of The Review:**

The idea of the paper is novel, interesting, and clearly exposed. However, a wealth of clarity issues in the experiments section hinder the clarity of the paper overall.

---

> ### Author Response · Authors · 2021-11-22
> **Response to the Reviewer**
>
> We thank the reviewer for the helpful comments. We have revised the manuscript to address the comments. The responses are listed below.
>
> Comment 1: Regarding the plots of the result
>
> Respone : We have presented new figures with much less clutter. We have used new color schemes and line types. The legend schemes are now explained in detail in the figure captions. Refer Figures 4 and 5 in the revised manuscript.
>
>
> Comment 2: The loss of the ramanujan property should be present on the testing accuracy graphs.
>
> Response: We have now shown the accuracy and Ramanujan graph properties in the same plots. Refer Figures 4 and 5.
>
>
> Comment 3: Experimental results: "in some cases (e.g. below Figure 1), switches between the proportion of edges pruned and the proportion of edges kept"
>
> Response: We made these parameters uniform over text and all the figures.
>
>
> Comment 4: I understand why you wouldn't want a rigorous definition of a universal cover, but you should add a reference
>
> Response: We have added a reference to Hoory et al 2006 Section 6, which describes universal cover.
>
>
> Comment 5: The theory of Ramanujan graphs is only stated for positive weights, yet this is not the case for neural networks: how is the ramanujan bound defined?
>
> Response: Since we are primarily interested in the connectivity properties of the neural networks we consider only the magnitude of the weights. Although the sign is important for the functional properties of the network, we have ignored the signs while analysing the structural properties
>
>
> Comment 6: Typo below def. 3: it is the Alon-Boppana theorem
>
> Response: We have corrected the typo.

---

### Author Response · Authors · 2021-11-22
**Overview of the revision**

We thank the reviewers for the helpful comments. We have revised the manuscript to address the comments. Major changes made are:

We now highlight the contributions of the work.

The primary contribution of the paper is the empirical demonstration that winning tickets often satisfy the Ramanujan graph property. Based on these observations we propose a modification to iterative pruning algorithms which attempts to preserve Ramanujan property for as many layers as possible by using the spectral bound criteria to determine the layers which are still amenable to pruning.

We have revised the experimental results section. Many of the plots are reorganized and combined. We hope that the new plots will better
explain the claims regarding the connection between the lottery ticket hypothesis and Ramanujan graph properties of the network.

We have included the results comparing the modified Ramanujan-Bound based pruning criteria with the state-of-the-art pruning algorithms.

---

> ### Author Response · Authors · 2021-11-30
> **Broad summary of the revision**
>
> The comments of the reviewers helped us to make significant revisions of the manuscript. A number of new experiments are performed. The experimental results section is reorganized. Presentations of the plots are changed. We hope that the contribution of the work is highlighted in the new version. The following claims are made based on experiments.
>
> Our primary contribution is the empirical observation that `winning tickets' often satisfy the Ramanujan graph property. Ramanujan graphs are optimally sparse graphs that maintain resilient connectivity or the expander property. We believe that this is an important parameter to characterize information flow in a deep neural network.
>
> The property can be tested for individual  network layers by considering bounds on the spectral properties of the corresponding bipartite graphs (Table 1). We present a small example in Figure 2 to illustrate the fact that if individual layers satisfy the Ramanujan graph property; then the entire network maintains connectivity. Also, computing the bounds for individual layers does not have a large computational overhead.
>
> We perform extensive experimental studies of LeNet, Conv4 architectures on MNIST and CIFAR10 to demonstrate the above phenomenon. The salient results are presented in Figures 4 and 5. More detailed results are presented in the Appendix. In our experiment, IMP pruning is used to obtain networks with reducing densities (i.e., remaining weights) using exactly the same mechanism followed by Frankle and Carbin 2019. As in Frankle and Carbin we observe that the test set accuracy does not reduce significantly till a certain density, and then starts declining steadily. Figures 4 and 5 shows that the densities from which the accuracy starts declining also marks the point from which the network layers begin to lose their Ramanujan graph property. However, as in the study of Frankle and Carbin the decline in accuracy is in many cases not sharp. We have also performed studies for noisy test examples at various Gaussian noise levels.
>
>
> Based on the above observations we attempt to modify existing iterative pruning algorithms by considering Ramanujan graph properties. In our modification of iterative pruning algorithms (Algorithm 1) we check Ramanujan graph properties of individual layers at each pruning iteration. We stop pruning at layers that have already lost the Ramanujan property and continue pruning in those that are still redundantly connected. Use of such local stopping criteria instead of global ones like validation accuracy allows us to schedule pruning across layers at different pruning iterations. This may potentially help us obtain a sparser network that better preserves connectivity.
>
> We have found in our experiments that most sophisticated pruning algorithms like IMP and SynFlow generate networks that satisfy the Ramanujan graph property for most layers except for extremely low network densities. However, we have observed that among two networks having the same low network density the one that has lost Ramanujan property for a lesser number of layers has a significantly higher accuracy. The addition of the Ramanujan graph property improves the performance of the SynFlow and IMP algorithms.
>
> In summary, we propose that study of Ramanujan graph properties of (pruned) neural networks is an effective means of characterizing its connectivity property. Experimentally we demonstrate that the property has strong links with the accuracy of the network. The observation may be exploited to design competitive pruning algorithms that preserves connectivity even at low network densities.
>
> We agree that analysis based on other similar properties of the network is possible. However, this is the single most important characterization of a network connected to the free flow of information without any bottlenecks as supported by theoretical considerations.

---

### Decision · Program_Chairs · 2022-01-20

**Decision:**

Reject

**Comment:**

Dear Authors,

The paper was received nicely and discussed during the rebuttal period. However, the current consensus suggests the paper requires another round of revisions before it gets accepted.

In particular:

- There were still some gray areas regarding comparison to simple techniques. E.g., one reviewer raised the question how it compares to simply stopping based on validation accuracy for example. The reviewer was missing the justification why stopping at the loss of Ramanujan graph property is preferable in comparison to other criteria.
- Several reviewers found the general idea interesting, but all felt that more reasonings about the impact/insights/relationship of Ramanujan graph property with pruning need to be found to get accepted.
- Reviewers appreciate that the authors corrected many parts of the submission (see increased scores). However, reviewers felt that the paper requires more data/evidence to get accepted at this level, based on the discussions made during the rebuttal period.

Best AC